# Improving the Measurement of Characteristic Parameters for the Determination of GHG Emissions in the Semiconductor and Display Industries in Korea

**Bong-Jae Lee** * , **Soo-Young Yun, In-Kwon Jeong, Yujin Hwang, Jun-Hyeok Park and Jonghoon Kim**

Korea Testing and Research Institute (KTR), 98 Gyoyukwon-ro, Gwacheon-si 13810, Gyeonggi-do, Republic of Korea; syy927@ktr.or.kr (S.-Y.Y.); step519@ktr.or.kr (I.-K.J.); wls0348@ktr.or.kr (Y.H.); jh1224ss@ktr.or.kr (J.-H.P.); di1024@ktr.or.kr (J.K.)
* Correspondence: jae8076@ktr.or.kr; Tel.: +82-2-2092-4056

**Abstract:** Semiconductor and display industries in the Republic of Korea make up the global electronics market with some of the greatest potential for growth due to accelerated digital transformation. Greenhouse gases (GHGs) present in the Earth's atmosphere could trap heat and contribute to the greenhouse effect, leading to global warming and climate change, and it is important to note that while GHGs are naturally present in the atmosphere and play a crucial role in regulating the Earth's temperature, human activities have significantly increased their concentration, leading to accelerated global warming and climate change. Volatile fluorinated compounds (FCs), including perfluorocompounds (PFCs), hydrofluorocompounds (HFCs), $NF_3$, and $SF_6$, are potent long-standing greenhouse gases that are used and emitted by electronics during the manufacturing and display stages of semiconductors. In accordance with global climate change, GHG reduction has developed as a demand of the times, and the electronics industry has also made efforts to reduce GHG emissions in response. Until now, process emissions from the use of fluorinated greenhouse gases (F-GHGs) in various industries have been calculated according to the '06 IPCC G/L, and emission factors of '06 IPCC G/L have also been applied. However, the reduction and emission factors proposed in the IPCC G/L are values that do not reflect the latest and advanced reduction technologies in South Korean electronics, and national GHG emissions are overestimated. In this paper, by preparing accurate measurement methods for destruction removal efficiency (DRE), the use rate of gas ($U_i$), and b-product emission factors ($B_{by-product,\,i}$), which are characteristic parameters for estimating GHG Tier 3a emissions, we aim to increase the accuracy of GHG emissions by advancing emission factors that are unique to the semiconductor and display industries within the Republic of Korea.

**Keywords:** destruction removal efficiency (DRE); use rate of gas; by-product emission factor; semiconductor; display; electronics; greenhouse gas (GHG); abatement equipment; point of use; scrubber; carbon neutrality; characteristic parameters

## 1. Introduction

Semiconductor and display industries in the Republic of Korea are continuously increasing greenhouse gas (GHG) emissions due to the rapid growth in production volume [1–7]. In total, 60–70% of GHG emissions from the semiconductor and display industries are indirect emissions released through electricity consumption, but they are still unlikely to contribute to GHG reductions for carbon neutrality by improving process efficiency [7,8].

There is a lack of related studies for GHG assessment in Korea. With these issues, various industries used the default emission factor presented in the Intergovernmental Panel on Climate Change (IPCC) guidelines for national GHG inventories to date. The IPCC has recently recommended applying country-specific reduction and emission factors rather than default values [9]. When country-specific emission factors for each industry that

have recently been studied are developed and compared with the previously applied IPCC default values, results showing significant differences have been confirmed. By comparing these results, we found that domestic semiconductor and display industries overestimate GHG emissions using IPCC default factors. Therefore, recognizing the fact that GHG emission calculations have been overestimated, various studies are being conducted to develop country-specific reduction and emission factors [9–13]. Through these efforts, determining appropriate GHG emission factors that are suitable for the nation's conditions will be essential for domestic GHG forecasting and reduction strategy establishment. Currently, process emissions from fluorinated gases in the domestic semiconductor and display industries are calculated based on the '06 IPCC guidelines, and applied emission factors are also based on '06 IPCC default values [14,15]. However, reduction and emission factors presented in the IPCC guidelines have been quite conservative and have not reflected the latest advanced abatement technologies in the domestic semiconductor and display industries. As a result, GHG emissions from domestic semiconductor and display facilities and national GHG emissions are being overestimated. Accordingly, developing site-specific emission factors at the workplace level by directly measuring GHG emission calculation characteristic parameters is necessary; thus, reducing GHG emissions and carbon neutrality may be realized. Nevertheless, there is currently insufficient research on directly estimating the characteristic parameters of GHG emission calculation, but through measurement research on the characteristic parameters of GHG emission calculation, we proposed clear measurement methods for securing the reliability of GHG emissions over facilities in the semiconductor and display sectors, as well as destruction removal efficiency (DRE), the use rate of gas ($U_i$), and by-product emission factor ($B_{by\text{-}product, i}$).

The most recently published '19 IPCC refinement G/L presented Tier 1, 2, and 3 methods for estimating GHG emissions in electronics manufacturing, and this paper included Tier 2 and 3 methods, which are the most generalized methods. This paper also focused on the Tier 3a method, which is a direct self-measurement method based on facilities. The classification and difference values according to the Tiers 1, 2, and 3 were tabulated, as shown in Table S1. The Tier 2 method could be applied when activity data on the process gas were used by each company, divided into Tiers 2a, 2b, and 2c. Among them, the Tier 2a method showed methodology application only to the semiconductor sub-sector using the default emission factors presented in the '19 IPCC refinement G/L, regardless of process and wafer size. The use rate of gas ($U_i$) and emission factor for by-products generated from input gas ($B_{k, i}$) were used in the Tier 2a emission calculation formula, indicating the average value of the process for each wafer size [15]. Compared to the Tier 2a method, the Tier 2b method showed the same formula but, dissimilarly, it considered the wafer size of 200 mm or less or 300 mm size. As a result, the method used for estimating GHG emissions based on wafer size is the most noticeable change in the semiconductor and display sectors of the '19 IPCC Refinement guidelines. The Tier 2c method was used for the sub-sector calculation of electronics, and both emission factors ($U_{i, p}$ and $B_{k, i, p}$) were provided as ($_p$) for each type of process [15,16]. The semiconductor process types were divided into six types: etching and wafer cleaning (EWC), remote plasma cleaning (RPC), in situ plasma cleaning (IPC), in situ thermal cleaning (ITC), $N_2O$ TFD, and $N_2O$ 'Other', etc., and display process types could be classified into four types: etching, remote plasma cleaning (RPC), in situ plasma cleaning (IPC), and $N_2O$ TFD [17]. In the case of the semiconductor process, the emission was obtained for each wafer size, but in the case of the display, the wafer size was separately ignored. When the default emission factors were not applicable, a ($1\text{-}U_i$) value of 0.8 was used, and the by-product factor value applied 0.15 for $CF_4$ and 0.05 for $C_2F_6$ [18].

In the case of the Tier 3a methodology, which is the focus of this study, the same calculation formula as Tier 2c was applied, but all coefficients ($U_{i, p}$, $D_{k, p}$, and $B_{k, I, p}$) required for calculation used values that were estimated by measuring each manufacturing process. Since the coefficient was calculated through direct measurement, it was also possible to estimate the emissions of the manufacturing process, equipped with a wafer

size of 450 mm, and research on accurate measurement methods for the use rate of gas ($U_i$), destruction removal efficiency (DRE), and by-product emission factor ($B_{by\text{-}product,\,i}$) was essential in order to accurately estimate GHG emissions. Furthermore, the Tier 3b method was added to Tier 3b to calculate emissions using the stack system, which measures the GHGs emitted by each process at the end-of-pipe (stack) process. Since this methodology also estimated GHG emissions by adopting the direct measurement method, it was also possible to measure emissions on the 450 mm wafer size and calculate GHG emissions by applying more variables [15].

In this paper, by offering accurate measurement methods for the destruction removal efficiency (DRE), use rate of gas ($U_i$), and by-product emission factor ($B_{by\text{-}product,\,i}$), which are characteristic parameters for calculating GHG emissions of the Tier 3a method, we aimed to increase the accuracy of GHG emissions by improving emission factors that are unique to the semiconductor and display industries of the Republic of Korea.

## 2. Materials and Methods

### 2.1. Measurement Methods and Conditions of the DRE, Use Rate of Gas ($U_i$), and By-Product Emission Factor ($B_{by\text{-}product,\,i}$)

DRE, the use rate of gas ($U_i$), and by-product emission factor ($B_{by\text{-}product,\,i}$) were calculated on site using the semiconductor and display manufacturing facilities during normal operation. Based on the on-site circumstances and process gas being used, the volume flowrate of the gas entering and gas emitting processes from the abatement equipment was measured using a quadrupole mass spectrometer (QMS), and the concentration of the targeted FCs or $N_2O$ was measured using a Fourier transform infrared spectrometer (FT-IR) (Figure 1).

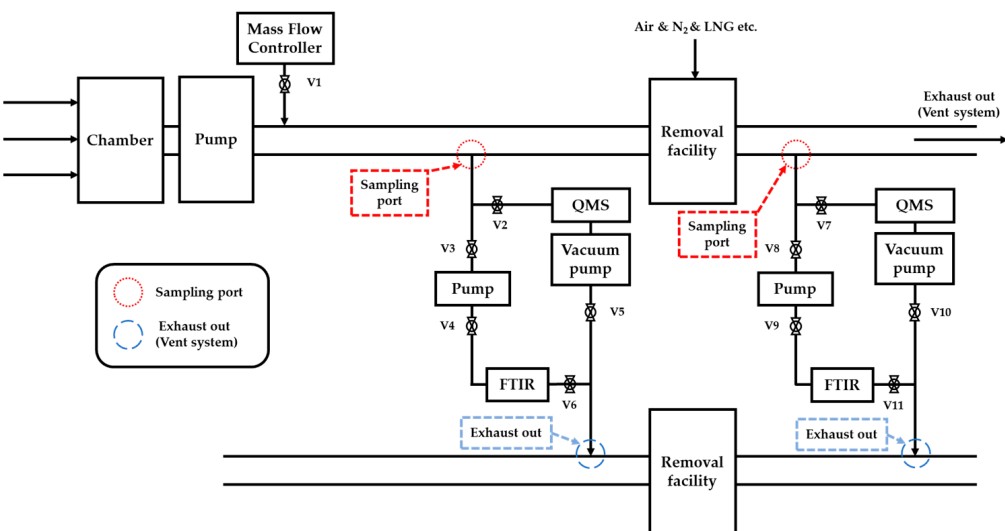

**Figure 1.** Diagram of measuring device installation for measuring DRE.

The QMS and FT-IR equipment regulations used for measurement purposes are specified in SEMATECH [16], and measurements were performed using the equipment that complies with them. QMS is a mass spectrometer, mainly used for gas analysis, that uses an ion separation quadrupole. In this study, it was used for the purpose of calculating the flow rate of process exhaust gas by measuring the concentration of the injected tracer gas and calculating the dilution ratio, and a mass analyzer with a minimum specification of 0 to 100 amu should be used to include both the composition of the exhaust gas and the mass range of the tracer gas. Additionally, in order to maintain the vacuum pressure, a vacuum pump and pressure gauge were installed and used together. FT-IR is a device that enables qualitative and quantitative analysis by measuring the number of infrared rays absorbed by a sample, and it is used for the purpose of measuring the concentration of the target substance contained in the exhaust

gas from the process. The gas cell installed in the FT-IR uses standardized cells and vacuum-only fittings, and is accurately placed on the holder by attaching the window made of KBr, ZnSe, or Ge/ZnSe material through which infrared light passes. The gas cell that is suitable for the type and concentration of the target gas should be applied, as well as the length for each type of scrubber, as mentioned in Table S3.

This measurement guidelines used for estimating the DRE, use rate of gas ($U_i$), and by-product emission factor ($B_{by\text{-}product, i}$) apply nitrous oxide ($N_2O$), hydrofluorocarbon (HFC), perfluorocarbon (PFC), sulfur hexafluoride ($SF_6$), and nitrogen trifluoride ($NF_3$), among other semiconductors and display manufacturing process emissions as target substances in order to calculate the GHG emissions presented in the guidelines for reporting and certifying the trading of GHG emissions. Hydrofluorocarbons (such as HFC-23 ($CHF_3$) and HFC-32 ($CH_2F_2$)) and perfluorocarbons (such as PFC-14 ($CF_4$), PFC-116 ($C_2F_6$), PFC-218 ($C_3F_8$), and PFC-c318 ($c\text{-}C_4F_8$)) were used in this measurement method, and the detection limit (DL) for each gas cell of FT-IR was in accordance with Table S2.

Interfering substances that were found to affect the measurement's results in the process included moisture found via spectrum interference. In order to eliminate this interference, the gas sample must be heated to 100 °C or more before the gas sample flows into the analysis device to minimize the inflow of moisture and substances, causing interference. Due to overlapping with the reference spectrum region of the target gas, measurements should be reset to a spectrum that does not overlap in order to minimize interference.

The tracer gas injected into the pipeline to calculate the flow rate of the process exhaust gas was chemically stable and was also well mixed and diffused with the atmosphere, and the tracer gas that was applied used helium (He), neon (Ne), argon (Ar), krypton (Kr), and xenon (Xe) gases, which are inert gases used for measuring the volume flow rate of GHG emissions from the manufacturing process. The injecting position of the tracer gas was at the rear end of the pump in the manufacturing process and the distance between the inlet of the tracer gas and the inlet of the scrubber should be installed at least 1 m (POU scrubber) or 5 m (house scrubber) so that the tracer gas could be sufficiently mixed. Before measurements were taken, the whole part of the measuring instrument should be inspected. In particular, it is necessary to check whether gas leaked, and the power should be turned on according to the order, and the sample's standby flow rate and other conditions should be adjusted according to the manual. Then, when the steady state was reached, QMS and FT-IR instruments should be used to calibrate for accurate measurements. Finally, on-site manufacturing was facilitated during normal operation, and QMS and FT-IR operate to continuously estimate the volume flow rate, and concentration of the target gas for 1 h in order to calculate the DRE, use rate of gas ($U_i$), and by-product emission factors ($B_{by\text{-}product, i}$). These measurements were calculated at each scrubber point in semiconductor and display facilities.

### 2.2. Destruction Removal Efficiency (DRE)
#### 2.2.1. Characteristic Parameter Features

Destruction removal efficiency (DRE) refers to the ratio of the destroyed or reduced greenhouse gas emitted from the manufacturing process by the GHG abatement equipment, such as the point-of-use (POU) scrubber or house scrubber, which is located at the rear of the semiconductor, and the display manufacturing process chamber, revealing a significant difference in the GHG emissions depending on the DRE factor, which makes DRE the most critical factor in determining the GHG emissions in the semiconductor and display industries.

According to the 2019 IPCC Refinement, the DRE of fluorinated compound (FC) gas is 0.89~0.99 and the DRE of $N_2O$ gas is 0.6 (Table 1).

As it is the most critical factor in determining GHG emissions, various studies have been conducted, including SEMATECH, ISMI, UN EPA, KTR, etc., and a DRE measurement method using Fourier transform infrared spectroscopy (FT-IR) and quadruple mass spec-

trometry (QMS) have been established using an EPA protocol in USA and Korea Standard (KS) in the Republic of Korea [16–25].

**Table 1.** Default DRE factors for GHG emissions in semiconductor and display industries.

| Gas | DRE |
|---|---|
| $CF_4$ | 0.89 |
| $C_2F_6$ | 0.96 |
| $C_3F_8$ | 0.95 |
| $C_4F_6$ | 0.99 |
| c-$C_4F_8$ | 0.98 |
| $C_4F_8O$ | 0.98 |
| $C_5F_8$ | 0.98 |
| $CHF_3$ | 0.98 |
| $CH_2F_2$ | 0.98 |
| $CH_3F$ | 0.98 |
| $CH_2F_5$ | 0.95 |
| $NF_3$ | 0.95 |
| $SF_6$ | 0.95 |
| $N_2O$ | 0.60 |

2.2.2. Calculation of Destruction Removal Efficiency (DRE)

The average volume flow rate of the exhaust gas from the process can be calculated from the measured concentration of the tracer gas using the QMS based on Equation (1).

$$F = \frac{S_f}{C_{Kr} \times 10^{-6}} \tag{1}$$

F: the average volume flow rate of exhaust gas from the process based on a single concentration data points (L/min);
$S_f$: the volume flow rate of tracer gas injected using a MFC (L/min);
$C_{Kr}$: the measured concentration of tracer gas using the QMS (μmol/mol).

The average volume flow rate is calculated using Equations (2) and (3);

$$F_m = \sum_{1}^{n} \frac{F_i}{n} \tag{2}$$

$$\sigma_{Fm} = \sqrt{\frac{1}{n} \sum_{1}^{n} (F_i - F_m)^2} \tag{3}$$

$F_m$: the average volume flow rate of tracer gas from 1~$n$ times measurements (L/min);
$F_i$: the measured volume flow rate of process emission gas from $i$ time measurements (L/min);
$n$: number of measurements;
$\sigma_{Fm}$: relative errors.

The volume flow rate of FCs can be determined based on the concentration of GHGs that are measured with FT-IR (Equation (4)).

$$V_i = \sum_{i=1}^{N} F_{m,i} C_{i,j} = F_{m,i} \sum_{i=1}^{N} C_{i,j} \tag{4}$$

$V_i$: the volume flow rate of FC gas $i$ (L/min);
$F_m$: the average volume flow rate of tracer gas from 1~$n$ times the number of measurements (L/min);
$C_i$: the concentration of FC gas $i$ (μmol/mol);
$C_{i,j}$: the concentration of FC gas $i$, $j$ entering, or being emitted from, the abatement equipment (μmol/mol);

*j*: the concentration of gas entering, or being emitted from, the abatement equipment.

Using the volume flow rate of the entering and emitting FCs from Equation (4), the DRE is determined using Equation (5).

$$\text{DRE} = \left(1 - \frac{V_{i,out}}{V_{i,\,in}}\right) \times 100(\%) \tag{5}$$

$V_{i,in}$: the volume flow rate of FC gas *i* that flows into the abatement equipment per unit time under normal operating conditions (L/min);
$V_{i,out}$: the volume flow rate of FC gas *i* that flows out of the abatement equipment per unit of time under normal operating conditions (L/min).

### 2.2.3. Improving DRE Measurements

To measure the DRE of the semiconductor and display manufacturing processes, the volume flow rate of the process gas was calculated by estimating the concentration of tracer gas, as mentioned in Section 2.2.2. Tracer gas is generally applied as an inert gas that interacts with other gases in the pipe and is not destroyed by the abatement system, and Kr (Krypton) gas is used among several inert gases because it must be applied as a gas that is not used in semiconductor and display manufacturing processes. Accordingly, tracer gas (Kr) is injected using a mass flow controller (MFC) at the front of the abatement system. However, an MFC is generally calibrated using nitrogen or compressed air and when an MFC is calibrated by nitrogen or when compressed air is used, the accuracy of the volume flow rate of Kr is decreased. Therefore, accurate measurements were only possible when the volume flow rate of Kr was corrected by applying the gas factor, as shown in Table 2, and the composition of internal process gas in the semiconductor and display industry is shown in Table S4.

**Table 2.** Gas factors in nitrogen calibration standard using inert gas.

| Inert Gas | Gas Factors in Nitrogen Calibration Standard |
|---|---|
| He | 1.386 |
| Ne | 1.398 |
| Kr | 1.382 |
| Xe | 1.383 |

Accurate measurements of the QMS for estimating the volume flow rate of process gas are only possible when samples are manufactured and equipment is calibrated, similar to the configuration of the actual internal process gas. Accordingly, a mixed standard gas similar to the composition of the process gas is produced, and the Kr concentration to be measured is diluted with $N_2$, and then the true estimation values are calculated in Table 3 to draw comparisons with the measured values.

When Kr (99.999%) was injected into 1 SLM by estimating the volume flow rate of the process gas through the concentration measurement of Kr, measured concentrations were used to calculate the volume flow rate, according to Equation (1). In Experiment 1, when the volume flow rate was calculated, after calibration in consideration of only Kr without the composition of the process gas, the estimated true error value rate was 1.8 to 4.37% (Figure 2 and Table 4).

In Experiment 2, when all components of the mixed standard gas were calibrated to measure the Kr concentration, in light of the composition of the process gas, a normalization process was performed to estimate the volume flow rate of the process gas, and the error rate with true values was 0.74 to 2.26% (Table 5).

**Table 3.** Composition of prepared mixed standard gas and comparison with measured values through dilution of $N_2$.

| Volume Flow Rate ($N_2$, SLM) | | Volume Flow Rate (Standard, SLM) | | Concentration (%) | |
|---|---|---|---|---|---|
| 5 | | 5 | | 1.01 | |
| 3 | | 7 | | 1.414 | |
| **Mixed Standard Gas** | | | | | |
| Kr | $N_2$ | $O_2$ | Ar | $CO_2$ | Cylinder No. |
| 2.02 | 87.75 | 9.01 | 0.60 | 0.42 | N1030 |

Internal temperature in lab: −20 °C. MFC—calibrated product using compressed air. Assumption for volume flow rate calculation of injected Kr (99.999%) gas: 1 SLM.

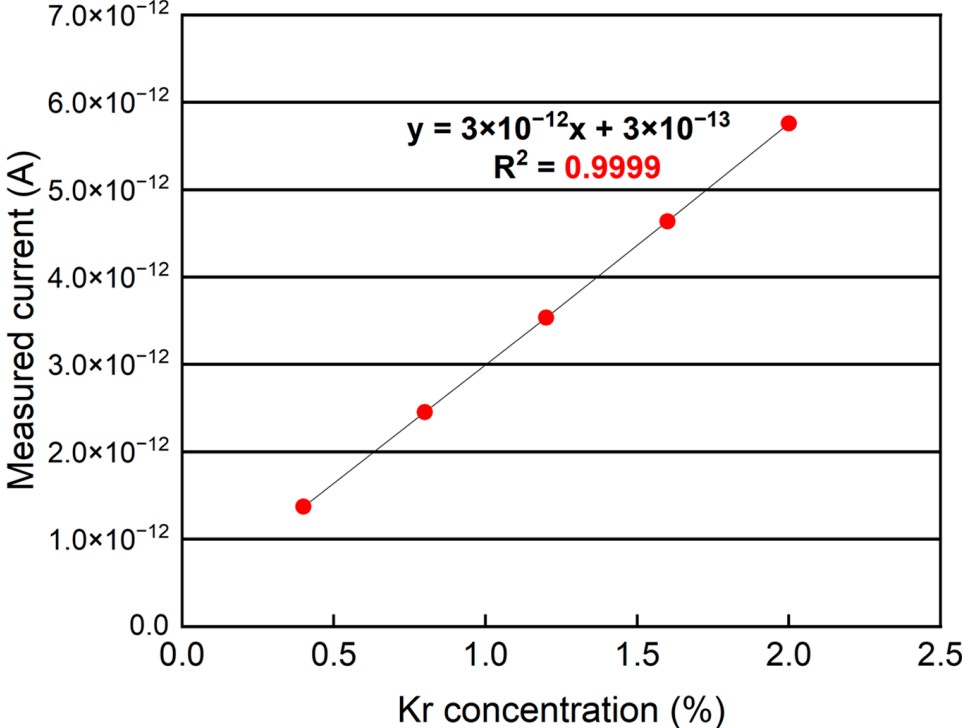

**Figure 2.** The error rate graph comparing true values using tracer gas (Kr) (Experiment 1).

**Table 4.** Arithmetic error rate values compared to true values in Experiment 1.

| $N_2$ | STD Gas | Pre-Set Kr Concentration (%) | Measured Kr Concentration (%) | Gas Factor | Temperature and Pressure Calibration | Measured Volume Flow Rate (SLM) | Calculated True Value of Volume Flow Rate | Error Rate (%) |
|---|---|---|---|---|---|---|---|---|
| 7 | 3 | 0.6 | 0.627 | 1.382 | 1.073 | 236.413 | 247.2 | 4.37 |
| 5 | 5 | 1.0 | 1.027 | 1.382 | 1.073 | 144.369 | 148.3 | 2.67 |
| 3 | 7 | 1.4 | 1.425 | 1.382 | 1.073 | 104.036 | 105.9 | 1.80 |

Experiments 1 and 2 confirmed that the concentration of Kr and the volume flow rate of process gas calculated through calibration based on Kr alone were accurately measured compared to the volume flow rate of the process gas. In addition, in order to confirm the accuracy of measurements, the estimated true value was analyzed by adding He gas to the mixed standard gas, as shown in Experiment 3, and, as the injected concentration of He gas increased, the normalization rate was found to also increase (Table 6 and Table S3 and Figure 3).

**Table 5.** Arithmetic error rate values compared true values in Experiment 2.

| N$_2$ | STD Gas | Pre-Set Kr Concentration (%) | Measured Kr Concentration (%) | Gas Factor | Temperature and Pressure Calibration | Measured Volume Flow Rate (SLM) | Calculated True Value of Volume Flow Rate | Error Rate (%) |
|---|---|---|---|---|---|---|---|---|
| 7 | 3 | 0.6 | 0.613 | 1.382 | 1.073 | 241.612 | 247.2 | 2.26 |
| 5 | 5 | 1.0 | 1.01 | 1.382 | 1.073 | 146.493 | 148.3 | 1.23 |
| 3 | 7 | 1.4 | 1.410 | 1.382 | 1.073 | 105.166 | 105.9 | 0.74 |

**Table 6.** Injected He gas concentration variations and their corresponding error rates.

| He Gas Concentration (%) | Error Rate (%) |
|---|---|
| 67.46 | 28.62 |
| 58.02 | 36.92 |
| 47.95 | 45.78 |
| 37.20 | 55.24 |
| 25.68 | 65.37 |
| 13.31 | 76.24 |

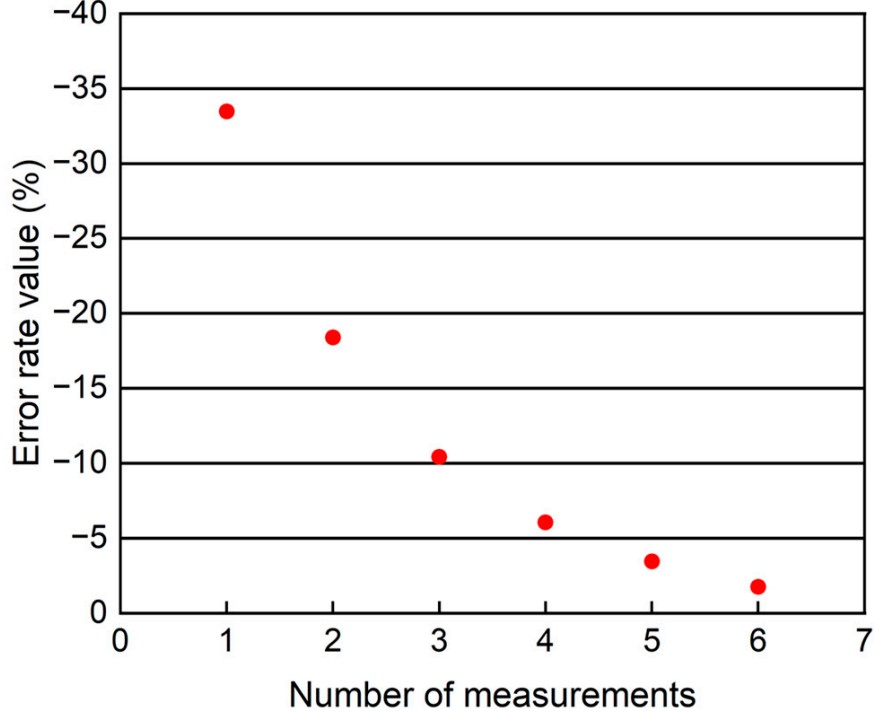

**Figure 3.** Error rate graph for comparing injected He concentration variations.

As a result, when performing the normalization process, considering unknown substances for accurate measurements, it is necessary to estimate the production of mixed standard gas after calibration by reflecting at least 85% of the process gas composition. It should be noted that when it is not reflected, the accuracy should be significantly lower than that of standard Kr materials manufactured from N$_2$-based standard gas.

2.2.4. Validation of Improved Measurement

To confirm the validity of the improved measurement method, linearity, accuracy, precision, and verification and quantification limits were analyzed using CHF$_3$ and C$_2$F$_6$. At first, to confirm linearity, measurements of 5-point concentration ranges (20, 40, 60, 80, and 100 ppm) of CHF$_3$ and C$_2$F$_6$ were taken using certified reference material (CRM) of 100 ppm N$_2$ balance. As a result, the correlation coefficient (R$^2$) values of CHF$_3$ and C$_2$F$_6$ confirmed that R$^2$ values exceeded 99% or more (Figure 4).

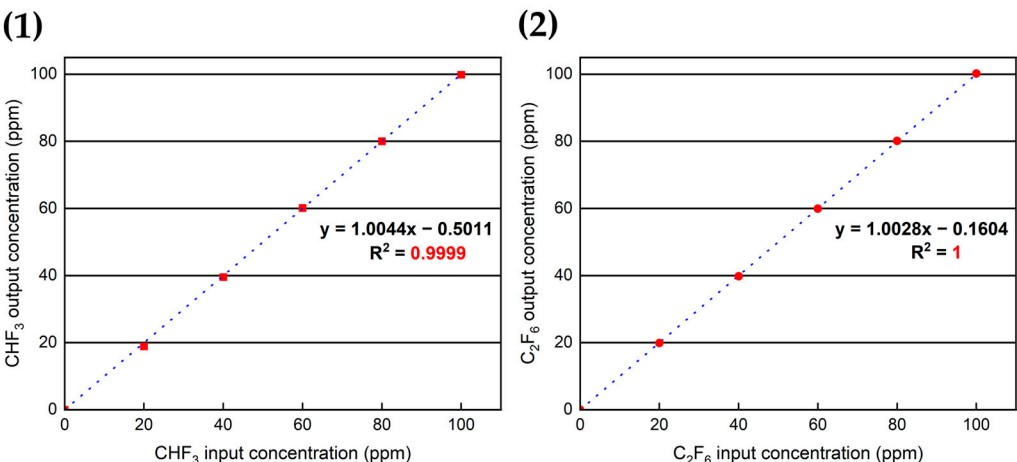

**Figure 4.** The linearity result graphs of (**1**) $CHF_3$ and (**2**) $C_2F_6$ to confirm validity.

Then, to confirm accuracy, the standard material of the median concentration for each item was analyzed four times using a CRM, and the average measurement value was divided by the median concentration value to obtain the results. This analysis was repeated 5 times with 60 ppm, which is the median concentration of CRM (100 ppm) from $CHF_3$ and $C_2F_6$. Then, the average repeated analysis values were divided by the median concentration values (60 ppm) and calculated as a percentage. As a result, the accuracy values of $CHF_3$ and $C_2F_6$ were calculated as 100.09 and 99.89%, respectively, showing that the accuracy of $CHF_3$ and $C_2F_6$ was estimated within a validity range of 90–110%.

To confirm precision, repeatability and reproducibility were verified. Repeatability was determined by analyzing the standard material of the median concentration for each item 4 times using a standard gas, and calculating the relative standard deviation (RSD). In order to confirm the repeatability, the analysis was repeated 5 times with 60 ppm, which is the median concentration of CRM (100 ppm) of $CHF_3$ and $C_2F_6$, and the RSD of analysis values were estimated. As a result, the RSDs of $CHF_3$ and $C_2F_6$ were calculated as 0.07 and 0.03%, and the values were confirmed to be within a validity range of 5%.

Reproducibility was calculated over a 5-point concentration range using the CRM (100 ppm) of $CHF_3$ and $C_2F_6$ by measuring four or more analyses from the testing laboratory and by performing analysis at different times. As a result, the accuracy measured to check the reproducibility of $CHF_3$ and $C_2F_6$ was analyzed within the validity range of 90 to 110%.

Furthermore, the detection limit was analyzed based on a 500 cm gas cell. The detection limit was calculated by repeatedly measuring the concentration (that is twice that of the manufacturer's provided detection limit) 7 times, and then multiplying the standard deviation by 3.14 to calculate the expanded uncertainty. The measurement equipment's detection limits of $CHF_3$ and $C_2F_6$ provided by the manufacturer were 1.5 and 0.5 ppm, and were calculated by multiplying the standard deviation of the values (3.14) measured 7 times at 3 ppm and 1 ppm, which are double the concentration values. As a result, the detection limit values of $CHF_3$ and $C_2F_6$ were calculated as 1.00 and 0.55 μmol/mol.

In addition, the quantitative limit was calculated as 10 times that of the detection limit, and the quantitative limit values of $CHF_3$ and $C_2H_6$ were estimated to be 3.19 and 1.75 μmol/mol. Overall, the validity of the improved measurement method was proven to be satisfied within the validity standard value range (Tables S6 and S7).

### 2.3. Use Rate of Gas ($U_i$)

2.3.1. Features of Characteristic Parameter

The use rate of gas ($U_i$) is the fraction of the injected GHG destroyed or transformed in the manufacturing process, such as etching and CVD. The use rate of gas can be determined depending on how optimized the process is. According to the 2019 IPCC

Refinement, the use rate of the fluorinated gas varies on the gas type, process type, and wafer size (Tables S8 and S9).

2.3.2. Research on Improving the Use Rate of Gas ($U_i$) Measurements

In the case of measuring the use rate of gas ($U_i$), no research method or standardized method has been proposed. Accordingly, this study proposes various methods that can be used to measure the use rate of gas ($U_i$) and compare the measurement results.

The first method of measuring the use rate of gas ($U_i$) involves installing a sampling port at the inlet of the point-of-use (POU) scrubber and measure the FCs or $N_2O$ gas volume flow rate in the plasma (on/off state) of the main process using measurement equipment (QMS and FT-IR), as shown in Figure 5.

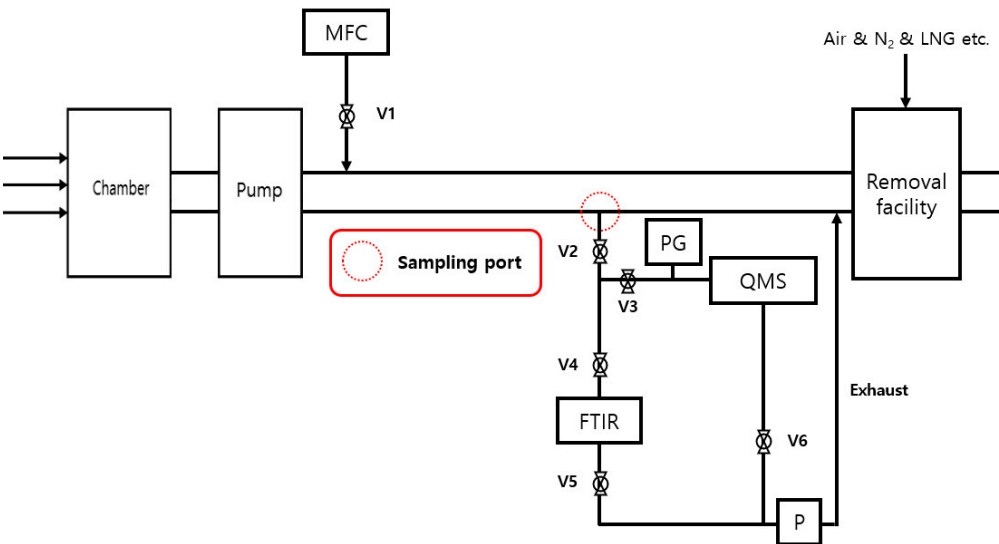

**Figure 5.** Diagram of measuring device installation for measuring the use rate of gas ($U_i$) [Method 1].

$$F_{on,off} = \frac{S_{f(on,off)}}{C_{Kr(on,off)} \times 10^{-6}}$$
(6)

$F_{on,off}$: the inlet volume flow rate at on/off plasma (L/min);
$S_{f(on,off)}$: the volume flow rate of tracer gas injected using an MFC at plasma (on/off state) (L/min);
$C_{Kr(on,off)}$: the measured concentration of the tracer gas using the QMS at plasma (on/off state) (μmol/mol).

The volume flow rate of FCs or $N_2O$ gas is calculated from the estimated inlet volume flow rate.

$$V_{on,off} = C_{(on,off)} \times F_{on,off} \times 1000$$
(7)

$V_{on,off}$: the inlet volume flow rate of FCs or $N_2O$ gas at plasma (on/off state) (sccm);
$C_{(on,off)}$: the measured concentration of FCs or $N_2O$ gas using the QMS at plasma (on/off state) (μmol/mol).

Therefore, the use rate of gas ($U_i$) is determined using Equation (8).

$$U_i = \frac{V_{off} - V_{on}}{V_{off}} \times 100 \ (\%)$$
(8)

$U_i$: the use rate of gas (%);
$V_{off}$: FC or $N_2O$ gas volume flow rate of plasma (off state) in the process chamber (sccm);
$V_{on}$: FC or $N_2O$ gas volume flow rate of plasma (on state) in the process chamber (sccm).

The second method of measuring the use rate of gas ($U_i$) involves calculating and comparing the volume flow rate of the chamber using an MFC with FCs or $N_2O$ gas volume flow rate measured at the inlet of the removal facility (POU scrubber) during the operating process (Figure 6).

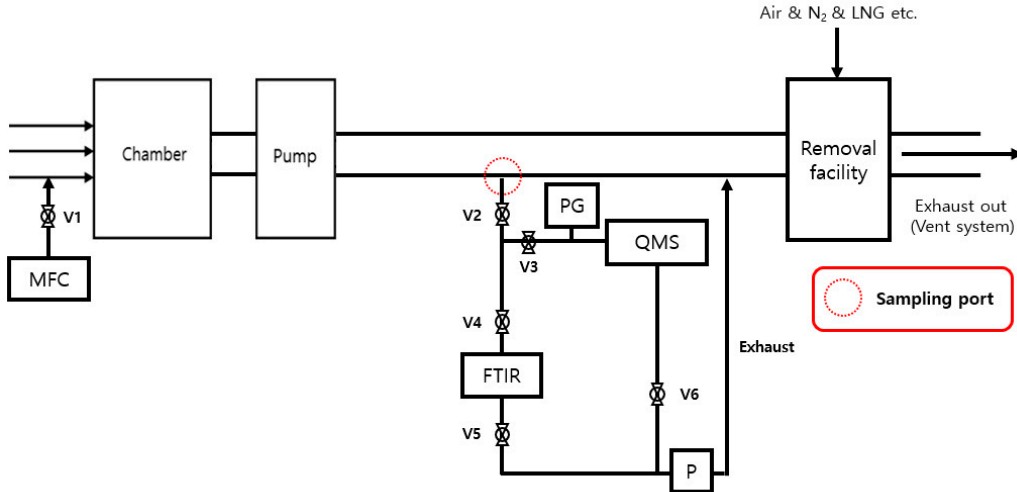

**Figure 6.** Diagram of measuring device installation for measuring the use rate of gas ($U_i$) [Method 2].

$$F = \frac{S_f}{C_{Kr} \times 10^{-6}} \tag{9}$$

$F$: the inlet volume flow rate (L/min);
$S_f$: the volume flow rate of tracer gas injected using an MFC (L/min);
$C_{Kr}$: the measured concentration of tracer gas using the QMS (μmol/mol).

The volume flow rate of GHG emissions at the inlet is estimated from the calculated inlet volume flow rate.

$$V_{in} = C \times F \times 1000 \tag{10}$$

$V_{in}$: the volume flow rate of FCs or $N_2O$ gas at the inlet (sccm);
$C$: the measured concentration of FCs or $N_2O$ gas at the inlet (μmol/mol).

Therefore, the use rate of gas ($U_i$) can be determined using Equation (11).

$$U_i = \frac{V_s - V_{in}}{V_s} \times 100 \ (\%) \tag{11}$$

$U_i$: the use rate of gas (%);
$V_s$: FCs or the $N_2O$ gas volume flow rate of GHGs from the main process (sccm);
$V_{in}$: FCs or the $N_2O$ gas volume flow rate of GHGs at the inlet (sccm).

The non-steady state process in Method 1 is approximately equal to the actual process gas supplement in Method 2. Therefore, Method 1 can be applied when the amount of GHGs supplied during the measurement time in the facility's main process is provided and verified, and Method 2 can be applied when the number of FCs or the amount of $N_2O$ gas is impossible to provide.

### 2.3.3. Improved Measurement Validation

The validation and use rates ($U_i$) were confirmed simultaneously through an evaluation of the previously mentioned DRE measurements. It was also applied to the use rate of gas ($U_i$) and by-product emission factor ($B_{by\text{-}product,\ i}$) measurements, and the results mentioned in Section 2.2.4 were then verified.

*2.4. By-Product Emission Factor ($B_{by-product, i}$)*

2.4.1. Characteristic Parameter Features

The by-product emission factor ($B_{by-product, i}$) refers to the rate at which GHGs, that are injected into semiconductor and display manufacturing industries, are converted into a by-product, i.e., another GHG, via plasma operation, such as etching and CVD processes. According to the '19 IPCC Refinement G/L, the by-product emission factor ($B_{by-product, i}$) of fluorinated gas varies depending on the process gas and the by-product gas, as shown in Tables S10–S15.

2.4.2. Research on Improving By-Product Emission Factor ($B_{by-product, i}$) Measurements

In the case of measuring the by-product emission factor ($B_{by-product, i}$), no research method or standardized method has been proposed. However, as mentioned in Section 2.3.2, measurements can be taken in the same way as the use rate of gas ($U_i$), showing differences in measuring the concentration of by-product gases instead of estimating the concentration of GHGs. The by-product emission factor ($B_{by-product, i}$) can be determined using Equations (12) and (13).

$$V_{by-pass} = C_{by-pass} \times F \times 1000 \qquad (12)$$

$V_{by-pass}$: the volume flow rate of by-product gas from the inlet of the abatement system (mL/min);
$C_{by-pass}$: the concentration of by-product gas as an FC from the abatement system (μmol/mol).

$$B_i = \frac{V_{by-pass}}{V_s} \times 100 \ (\%) \qquad (13)$$

$B_i$: the by-product emission factor during the manufacturing process ($B_{by-product, i}$);
$V_s$: the FC gas volume flow rate during the manufacturing process (mL/min).

2.4.3. Validation of Improved Measurements

The by-product emission factor ($B_{by-product, i}$) was also simultaneously validated through an evaluation of the DRE measurement method, as mentioned above, which was applied equally to the use rate of gas (Ui) and the by-product emission factor ($B_{by-product, i}$), and the results mentioned in Section 2.2.4 were then verified.

## 3. Results

*3.1. Destruction Removal Efficiency (DRE)*

3.1.1. Process of Measurement

To measure the efficiency of POU scrubbers used in the sectors, measurements were conducted on a plasma-type scrubber, and the key equipment used for estimation included two FT-IRs used to analyze GHG concentrations and two QMSs to analyze Kr gas concentrations. Four types of GHGs were measured: $SF_6$, $CF_4$, $NF_3$, and $N_2O$, and one type of Kr gas was used to calculate the volume flow rate of the inlet and the outlet. Then, FT-IR and QMS were calculated using two MFCs, secondary reference materials, and high-purity nitrogen (99.999%) gas as a reference of material-based gas (Figure 7).

The QMS was calibrated before measurements were taken in the field, and the reliability of FT-IR could be secured even when the data were calculated through post-calibration due to the nature of the equipment. The distance between the gas injection mini-chamber and inlet was more than 2 m, and QMS sampling and FT-IR sampling were performed by connecting 1/4-inch Teflon tubes to the inlet and outlet sampling ports. Before measurement were taken, in order to check the inside of the scrubber, 1 SLM of high-purity Kr gas and 100 SLM of $N_2$ gas were injected to confirm that there was no change in the volume flow rate at the inlet and outlet, and then, the measurements were carried out. Furthermore, in order to check the volume flow rate, 30 SLM of $N_2$ gas was additionally injected into the outlet, and the volume flow rates of the inlet and outlet were varied and estimated.

The target gases to be measured were $CH_4$ and $NF_3$, which were measured once each, and measurements were taken seven times in total. One measurement time was 10 min, and the internal purging of FT-IR was performed using $N_2$ gas at the end of each measurement.

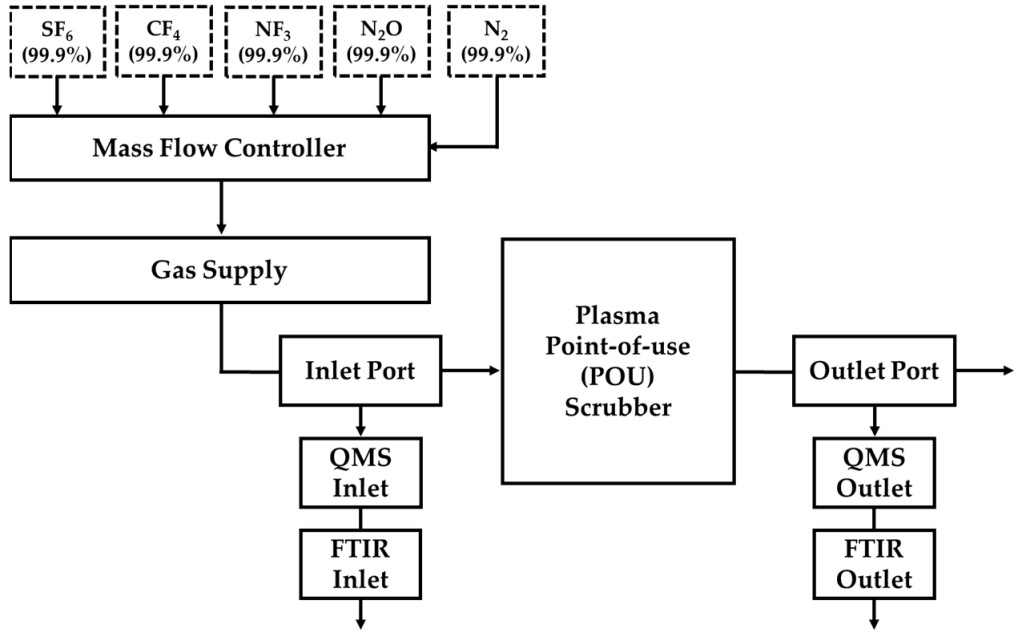

**Figure 7.** Diagram for measuring destruction removal efficiency (DRE).

3.1.2. Results of Measurement

In order to confirm the optimal conditions for each gas of the POU scrubber, the DRE for each volume flow rate and the consumption rates of plasma energy were measured. When the treatment volume flow rate of the POU scrubber increased to 300 SLM, it was confirmed that the DRE decreased to some extent. In particular, in the case of $NF_3$, it was necessary to increase the energy consumption for optimal DRE conditions, and 300 SLM was set as the applicable optimal processing capacity in display sectors. The measurement results are shown in Table 7. As an additional experiment, the DRE, according to the treatment capacity of the POU scrubber, was estimated based on $CH_4$, which was destroyed the least by the POU scrubber, and the measurement results are shown in Tables 7 and 8.

**Table 7.** Comparison of DRE values according to the volume flow rate variation in the POU scrubber based on the applicability to the display sector.

| Volume Flow Rate | Gas | Power Value (kW) | FT-IR (ppm) | | QMS (SLM) | | DRE (%) |
|---|---|---|---|---|---|---|---|
| | | | Inlet | Outlet | Inlet | Outlet | |
| 100 SLM | $SF_6$ | 7.095 | 5998.84 | 0.37 | 100 | 143 | 99.99 |
| | | 10.73 | 5998.84 | NA | 100 | 143 | 99.99 |
| | $NF_3$ | 6.058 | 4109.57 | NA | 100 | 143 | 99.99 |
| | | 10.73 | 4109.57 | NA | 100 | 143 | 99.99 |
| 300 SLM | $SF_6$ | 6.032 | 5414.71 | 163.84 | 322 | 365 | 96.57 |
| | | 10.73 | 5414.71 | NA | 322 | 365 | 99.99 |
| | $NF_3$ | 6.058 | 3786.96 | 1640.30 | 322 | 365 | 49.32 |
| | | 10.73 | 3786.96 | 143.33 | 322 | 365 | 95.57 |

**Table 8.** Comparison of DRE values according to volume flow rate variation in the POU scrubber based on the applicability to the semiconductor sector.

| Volume Flow Rate | Gas | Power Value (kW) | FT-IR (ppm) | | QMS (SLM) | | DRE (%) |
|---|---|---|---|---|---|---|---|
| | | | Inlet | Outlet | Inlet | Outlet | |
| 100 SLM | $CH_4$ | 8.08 | 5637.56 | 1086.10 | 100 | 143 | 72.45 |
| | | 9.22 | 5637.56 | 574.43 | 100 | 143 | 85.43 |
| | | 10.10 | 5637.56 | 275.43 | 100 | 143 | 93.01 |
| | | 10.73 | 5637.56 | 173.46 | 100 | 143 | 95.60 |
| 130 SLM | $CH_4$ | 10.10 | 5593.82 | 1254.10 | 130 | 175 | 69.77 |
| | | 10.73 | 5593.82 | 1067.57 | 130 | 175 | 74.31 |
| | | 12.104 | 5593.82 | 620.46 | 130 | 175 | 85.07 |
| | | 14.508 | 5593.82 | 96.75 | 130 | 175 | 97.67 |
| 150 SLM | $CH_4$ | 14.508 | 5565.99 | 385.24 | 150 | 190 | 91.23 |
| | | 15.12 | 5565.99 | 280.12 | 150 | 190 | 93.63 |
| | | 15.84 | 5565.99 | 149.45 | 150 | 190 | 96.60 |

### 3.2. Use Rate of Gas ($U_i$)

As mentioned in Section 2.2.2, using Methods 1 and 2 proposed in the research based on improving measurements for the use rate of gas ($U_i$). In the display sector, the use rate of gas ($U_i$) was estimated for the CVD process of $N_2O$ gases, whose usage has increased alongside the production of OLED. The measurement results are shown in Table 9.

**Table 9.** The use rate of gas ($U_i$) from $N_2O$ gas using the CVD process in the display industry.

| Method 1 | | | Method 2 | | |
|---|---|---|---|---|---|
| Plasma On | Plasma Off | Use Rate of Gas (%) | Injected Volume Flow Rate (sccm) | Plasma On | Use Rate of Gas (%) |
| Volume Flow Rate (sccm) | Volume Flow Rate (sccm) | | | Volume Flow Rate (sccm) | |
| 86,092 | 72,339 | 16.0 | 85,000 | 72,339 | 14.9 |
| 85,561 | 73,311 | 14.3 | 85,000 | 73,311 | 13.8 |

### 3.3. By-Product Emission Factors ($B_{by-product, i}$)

For the fluorinated greenhouse gases (F-GHGs) of $C_2$ or higher, which are widely used in semiconductor and display manufacturing processes, by-product emission factors ($B_{by-product, i}$) for each by-product gas that could be generated based on plasma were measured, and by-product emission factors ($B_{by-product, i}$) for each by-product gas that could be generated under various conditions are shown in Table 10.

**Table 10.** By-product gas emissions generated from a plasma scrubber.

| Gas | Power Value (kW) | FT-IR (ppm) | | DRE (%) | By-Product Gas | | |
|---|---|---|---|---|---|---|---|
| | | Inlet | Outlet | | $CF_4$ (ppm) | $C_2F_6$ (ppm) | $CHF_3$ (ppm) |
| $C_2F_6$ | 6.032 | 4273.38 | 1942.31 | 48.48 | 764.32 | - | - |
| | 8.08 | 4331.56 | 1599.53 | 58.14 | 1423.73 | - | - |
| | 10.73 | 4334.33 | 1117.95 | 70.76 | 2333.24 | - | - |
| $CHF_3$ | 6.032 | 5150.92 | 1707.59 | 62.40 | 576.91 | 109.88 | - |
| | 8.08 | 5150.92 | 1162.32 | 74.42 | 783.09 | 168.75 | - |
| | 10.73 | 5150.92 | 587.93 | 87.07 | 887.91 | 154.32 | - |
| $C_3F_8$ | 6.032 | 2946.14 | 1602.21 | 38.35 | 1397.89 | 27.13 | 10.03 |
| | 8.08 | 2946.14 | 985.54 | 62.08 | 2214.25 | 179.29 | 6.52 |
| | 10.73 | 2946.14 | 482.12 | 81.45 | 3256.02 | 487.92 | 2.24 |

### 3.4. Results on the DRE, Use Rate of Gas ($U_i$), and By-Product Emission Factor ($B_{by\text{-}product, i}$) Measurements

According to DRE measurement results, a comparison was performed with the '06 IPCC G/L and '19 IPCC Refinement to the '06 IPCC G/L for National Greenhouse Gas Inventories, as shown in Table 11. It was confirmed that DRE showed a large difference in volume flow rate, gas type, and energy consumption data in the manufacturing process. As a result, since DRE coefficients for each gas presented in the IPCC G/L considered and presented only the type of gas, it is necessary to develop coefficients based on various facility conditions in order to apply DRE that is suitable for the site.

**Table 11.** Comparison of DRE values between the presented IPCC guideline's factor values ('06 and '19) and measured DRE values in this study.

| Sector | Gas | Volume Flow Rate (SLM) | Power Value (kW) | DRE (%) | | |
| --- | --- | --- | --- | --- | --- | --- |
| | | | | This Study | '06 IPCC | '19 IPCC |
| Semiconductor | $CF_4$ | 100 | 8.08 | 72.45 | 90.00 | 89.00 |
| | | | 9.22 | 85.43 | | |
| | | | 10.10 | 93.01 | | |
| | | | 10.73 | 95.60 | | |
| | | 130 | 10.10 | 69.77 | | |
| | | | 10.73 | 74.31 | | |
| | | | 12.104 | 85.07 | | |
| | | | 14.508 | 97.67 | | |
| | | 150 | 14.508 | 91.23 | | |
| | | | 15.12 | 93.63 | | |
| | | | 15.84 | 96.60 | | |
| Display | $SF_6$ | 100 | 7.095 | 99.99 | 90.00 | 95.00 |
| | | | 10.73 | 99.99 | | |
| | | 300 | 6.032 | 96.57 | | |
| | | | 10.73 | 99.99 | | |
| | $NF_3$ | 100 | 6.032 | 99.99 | 95.00 | 95.00 |
| | | | 10.73 | 99.99 | | |
| | | 300 | 6.058 | 49.32 | | |
| | | | 10.73 | 95.57 | | |

According to results on the use rate of gas ($U_i$) measurements, the EPA 40 U.S. Code of Federal Regulation's (CFR) Part 98 was compared against the '19 IPCC Refinement to the '06 IPCC G/L for National Greenhouse Gas Inventories, as shown in Table 12. The use rate of gas showed that the recipes varied depending on the characteristics of each manufacturing process and the product being made. As a result, our analysis showed a large difference based on the input of optimal or excessive gases for each characteristic of product, and then it was found that it was necessary to develop coefficients based on these differences.

**Table 12.** Comparison of the use rate of gas ($U_i$) measurements between the presented EPA and '19 IPCC guidelines and the use rate of gas ($U_i$) measurements in this study.

| Sector | Gas | Use Rate of Gas (%) | | |
| --- | --- | --- | --- | --- |
| | | This Study | EPA | '19 IPCC |
| Display | N2O | 14.9 | 40.0 | 37.0 |
| | | 13.8 | | |

According to the results of by-product emission factor ($B_{by-product, i}$) measurements, the '06 IPCC G/L and '19 IPCC Refinements were compared to the '06 IPCC G/L for National Greenhouse Gas Inventories, as shown in Table 13. The by-product gas' generated rate showed a large difference based on gas type and the amount of energy assumption, similar to the DRE, and the by-product gas' generated rate presented in the IPCC G/L was presented as coefficients with various process conditions. We also found that it was necessary to improve coefficients based on various facility conditions.

**Table 13.** Comparison between by-product emission factor ($B_{by-product, i}$) measurements and IPCC guidelines' factor value measurements ('06 and '19) presented in this study.

| Sector | Gas | Power Value (kW) | By-Product Emission Factors ($B_{by-product, i}$) (Target Gas/By-Product Gas) | | |
| --- | --- | --- | --- | --- | --- |
| | | | $CF_4$ (%) | $C_2F_6$ (%) | $CHF_3$ (%) |
| Display | $C_2F_6$ | 6.032 | 17.65 | - | - |
| | | 8.08 | 32.87 | - | - |
| | | 10.73 | 53.87 | - | - |
| | | 6.032 | 11.20 | 2.13 | - |
| | $CHF_3$ | 8.08 | 15.20 | 3.28 | - |
| | | 10.73 | 17.24 | 3.00 | - |
| | | 6.032 | 47.45 | 0.92 | 0.34 |
| | $C_3F_8$ | 8.08 | 75.16 | 6.09 | 0.22 |
| | | 10.73 | 110.52 | 16.56 | 0.08 |

## 4. Conclusions

In summary, we presented accurate measurement methods centered around DRE, the use rate of gas ($U_i$), and by-product emission factors ($B_{by-product, i}$), which are characteristic parameters used for estimating GHG emissions in the semiconductor, and display industry and fluorine-based GHGs used in the electronics manufacturing processes were measured at Korea's various facilities. As a result, it was confirmed that GHG emissions and national GHG emissions are overestimated compared to the reduction and emission factor of '06 IPCC G/L and '19 IPCC G/L Refinement that are currently used to calculate GHG emissions in the Republic of Korea. Through this study, based on the established GHG emission calculation characteristic parameters in semiconductor and display industries, the need for re-establishment based on the facility's applicability of various coefficients presented in the EPA 40 U.S. Code of Federal Regulations (CFR) (part 98; Subpart I) was confirmed by comparing the '06 IPCC G/L and '19 IPCC Refinement to the '06 IPCC G/L for National Greenhouse Gas Inventories. Furthermore, by measuring the destruction removal efficiency (DRE), the use rate of gas ($U_i$), and the by-product emission factor ($B_{by-product, i}$) data for GHGs that are used in the manufacturing process in semiconductor and display industries, but not presented in IPCC G/L, the need for additional study to prevent any overestimation of GHG emissions in the Republic of Korea's semiconductor and display industries was made more apparent. Furthermore, we believe that reduction and emission factors that are unique to the Republic of Korea and that are more accurately established through additional measurements of GHGs used in semiconductor and display manufacturing processes in Korea, as well as research on characteristic parameters, can significantly help to reduce national GHG emissions, providing a good frame of references for countries that need to develop country-specific reduction and emission factors.

**Supplementary Materials:** The following supporting information can be downloaded at: https://www.mdpi.com/article/10.3390/app13158834/s1. Table S1: Composition of Tiers 1, 2, and 3 in the electronics industry based on emission sources using guidelines for estimating and reporting greenhouse gas (GHG) emissions. Table S2: The detection limits (DLs) of FT-IR for each gas cell. Table S3: The classification of applicable FT-IR gas cell lengths according to scrubber type. Table S4: The gas composition of internal process gas in the electronics manufacturing process. Table S5: Gas compositions according to He gas injections for Experiment 3. Table S6: Measurements on comparing the validity (CHF3) of results between analysts to confirm the precision (reproducibility). Table S7: Measurements on comparing the validity (C2F6) of results between analysts to confirm precision (reproducibility). Table S8: Default use rate factors for GHG emissions in the semiconductor industry. Table S9: Default use rate factors for GHG emissions in the display industry. Table S10: GHG default emission factors for by-product generation rates in the semiconductor industry (Tier 2a). Table S11: GHG default emission factors for by-product generation rates in the semiconductor industry (Tier 2b, wafer-size, i.e., less than 200mm). Table S12: GHG default emission factors for by-product generation rates in the semiconductor industry (Tier 2b, wafer-size, i.e., 300mm). Table S13: GHG default emission factors for by-product generation rates in the semiconductor industry (Tier 2c, wafer-size, i.e., less than 200mm). Table S14: GHG default emission factors for by-product generation rates in the semiconductor industry (Tier 2c, wafer-size, i.e., 300 mm). Table S15: GHG default emission factors for by-product generation rates in the display industry (Tier 2c).

**Author Contributions:** All authors contributed to the research presented in this study. Conceptualization and methodology, B.-J.L.; writing—original draft preparation, B.-J.L. and I.-K.J.; writing—review and editing, S.-Y.Y. and J.K.; analysis, I.-K.J., Y.H., and J.-H.P. All authors have read and agreed to the published version of the manuscript.

**Funding:** This study was funded by Korea Environment Industry and Technology Institute (KEITI) and Korea Ministry of Environment (MOE) (no. 2022003560008).

**Institutional Review Board Statement:** Not applicable.

**Informed Consent Statement:** Not applicable.

**Data Availability Statement:** Not applicable.

**Acknowledgments:** This work was financially supported by Korea Environment Industry and Technology Institute (KEITI) as "Climate Change R&D Project for New Climate Regime", and by the Korea Ministry of Environment (MOE).

**Conflicts of Interest:** The authors declare no conflict of interest.

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
