# Peer review of "Improving the Measurement of Characteristic Parameters for the Determination of GHG Emissions in the Semiconductor and Display Industries in Korea"

_applsci, doi:10.3390/app13158834_

Round 1

Reviewer 1 Report

Overall, the results presented in the manuscript look interesting. Although I do not have expertise in this field, I believe this work could be a good addition to the literature. Below are a few minor concerns/suggestions that could improve the manuscript- 

1. GHG in the abstract should be defined.

2. I think the title looks a bit longer. 

3. More references should be added showcasing the novelty.

4. I would suggest removing the tables from the conclusions. 

Looks good to me. 

Author Response

Response to Reviewer 1 Comments

Reviewer #1

Reviewer’s Comment: Overall, the results presented in the manuscript look interesting. Although I do not have expertise in this field, I believe this work could be a good addition to the literature. Below are a few minor concerns/suggestions that could improve the manuscript

Response : We appreciate your comments.

Specific Comments

Point 1: GHG in the abstract should be defined.

Response 1: Thank you for your comment. We have now added definement of GHG in the abstract.

(Page 1 - abstract) Semiconductor and display industry of Republic of Korea is one of the global electronics markets with great potential for growth due to accelerated digital transformation. Greenhouse gases (GHGs) present in the Earth’s atmosphere could trap heat and contribute to the greenhouse effect, leading to global warming and climate change, and it is important to note that while GHGs are naturally present in the atmosphere and play a crucial role in regulating the Earth’s temperature, human activities have significantly increased their concentration, leading to accelerated global warming and climate change. Volatile fluorinated compounds (FCs), including perfluorocom-pounds (PFCs), hydrofluorocompounds (HFCs), NF3, and SF6, are potent long-lived greenhouse gases used and emitted by the electronics during the manufacture of semiconductor and display.

Point 2: I think the title looks a bit longer

Response 2: Thank you for your suggestion. We have now changed the article’s title.

(Page 1 – Title) Improvement of the Characteristic Parameters Measurement for the Determination of GHG emissions in the electronics in Korea.

Point 3: More references should be added showcasing the novelty.

Response 3: Thank you for your thoughtful comment. We have added the reference to the text.

(Page 1,2 – introduction)

The semiconductor and display industries in Republic of Korea are continuously increasing greenhouse gas (GHG) emissions due to the rapid growth in production volume [1-7]. 60-70 percent of GHG emissions from the semiconductor & display industries are indirect emissions released through electricity consumption but it is still insufficient to contribute to GHG reduction for carbon neutrality by improving process efficiency [7, 8].

There is a lack of related studies for GHG assessment in Korea. With these issues, various industries used the default emission factor presented in the Intergovernmental Panel on Climate Change (IPCC) guidelines for national GHG inventories to date. The IPCC currently recommended applying country-specific reduction and emission factors rather than default values [9]. When the country-specific emission factors for each industry that have been studied recently are developed and compared with the previously applied IPCC default values, the results showing significantly difference were confirmed. Comparing these results, it reflected that the domestic semiconductor and display industries are overestimating GHG emissions by using IPCC default factors. Therefore, recognizing the fact calculation of GHG emissions is being overestimated, various studies are being conducted to develop the country-specific emission factors [9-13].

  1. Eui-Chan Jeon, Soojeong Myeong, Jae-Whan Sa, Jinsu Kim, Jae-Huk Jeong, Greenhouse gas emission factor development for coal-fired power plants in Korea, Applied Energy, Vol 87, 1, 205-210, 2010
  2. S. Lee, J. Kim, J. Lee & E.-C. Jeon, Development of CO2 emission factors from a large circulating fluidized bed boiler, Energy Sources, Part A: Recovery, Utilization and Environmental Effects, Vol.38, 9, 1262-1268, 2016
  3. Changsang Cho, Seongmin Kang, Minwook Kim, Yoonjung Hong, and Eui-chan Jeon, Uncertainty Analysis for the CH4 Emission Factor of Thermal Power Plant by Monte Carlo Simulation, Sustainability (Switzerland), Vol. 10, 10, 2018
  4. Seongmin Kang, Seonghum Cho, Joonyoung Roh, and Eui-chan Jeon, Analysis of Main Factors for CH4 Emission Factor Development in Manufacturing Industries and Construction Sector, Energies, Vol. 13, 5, 2020
  5. Seongmin Kang, Seong-Dong Kim, and Eui-chan Jeon, Emission Characteristics of Ammonia at Bituminous Coal Power Plant, Energies, Vol. 13, 7, 2020

Point 4: I would suggest removing the tables from the conclusions.

Response 4: Takeing your advice, we have now moved the tables from conclusions to results and then additional sessions were created to organize the contents.

(Page 15~16 – Results)

3.4. Results of DRE, Use rate of gas (Ui) and By-product Emission factors (Bby-product, i) measurement

According to the results of DRE measurement, comparison was performed with the 06’ IPCC G/L and 19’ IPCC Refinement to the ’06 IPCC G/L for National Greenhouse Gas Inventories as following Table 11. It was confirmed that DRE showed a large difference by volume-flow rate, gas type, and energy consumption in manufacturing process. As a result, since DRE coefficients for each gas presented in the IPCC G/L considered and presented only the type of gas, and it was necessary to develop coefficients considering various facility conditions in order to apply DRE suitable for the site.

Reviewer 2 Report

The paper can be improved by considering the following comments:

-suggest to tabulate the difference in tier 2a, 2b, 2c, 3a etc

-suggest to include legend for Figure 1

-how about validation for improvement for section 2.2 and 2.3?

-Figure 6 is quite blur

-Table 0 cannot be found

-Results for section 3.2 and 3.3 need more explanation

-suggest to have one section specially for comparison (instead of putting all in the conclusion)

-for conclusion, suggest to provide more detailed recommendation based on the limitation of current study

-suggest to mention the tables putting in the supplementary materials

Significantly needs proof reading. 

Author Response

Response to Reviewer 2 Comments

Reviewer #2

Reviewer’s Comment: The paper can be improved by considering the following comments.

Response : We appreciate your comments.

Specific Comments

Point 1: Suggest to tabulate the difference in Tier 2a, 2b, 2c, 3a, etc.

Response 1: Thank you for your comment. We have now tabulated about difference in Tier 1, 2a, 2b, 2c, 3a, 3b and added the phrase.

(Page 2 - introduction) The most recently published ’19 IPCC refinement G/L presented Tier 1, 2, 3 methods for estimating GHG emissions in the electronics manufacturing, and this paper included Tier 2 and 3 methods, which are the most generalized method, this paper was conducted focusing on Tier 3a method which is a direct measurement method for self-measurement by facilities. The classification and difference according to the Tier 1, 2 and 3 were tabu-lated as following Table S1.

Method

Emission factor

Explanation

Tier

Criteria for application of GHG emission factors

Tier 1

Default factor

The default factors specified in the certification guidelines or presented by Greenhouse Gas Information Center (GIR).

-

Default emission factors present in IPCC Guideline

Tier 2

Country-specific factor

Country-specific factors verified and published by the Greenhouse Gas Information Center (GIR).

2a

Typical electronics industry type (Semiconductor, Display, PV)

2b

Electronics industry process type (Etching, CVD) and wafer size (≤200mm, 300mm)

2c

Detailed electronics industry process type (EWC, RPC, IPC, TFD, etc.) and wafer size (≤200mm, 300mm)

Tier 3

Facility-specific factor

Emission factors developed and reported through self-measurement by facilities.

Included all emission factors developed and applied by facilities are not specified in the Tier 1,2 factors.

3a

Reporting facilities use site-specific emission factors

3b

Reporting facilities measure GHG emission at the stack level

Table S1. Composition of Tier 1,2 and 3 in electronics industry by emission source in guidelines for estimating and reporting Greenhouse gas (GHG) emissions

Point 2: Suggest to include legend for Figure 1.

Response 2: Thank you for your suggestion. We have now added session in Materials and Methods. In session 2.1, the measurement methods and conditions were described in detail. 

(Page 3,4 – Materials and Methods)

2.1. Measurement methods anc conditions of DRE, Use rate of gas (Ui), By-product emission factor (Bby-product, i)

Calculation of DRE, Use rate of gas (Ui), By-product emission factor (Bby-product, i) is measured on-site at the semiconductor & display manufacturing facilities during normal operation. Considering the on-site circumstances and process gas being used, the volume-flowrate of the process gas entering and emitting from the abatement equipment is measured using quadrupole mass spectrometer (QMS), and the concentration of the targeted FCs or N2O, is measured using fourier transform infra-red spectrometer (FT-IR) (Figure 1).

The QMS and FT-IR equipment regulations used for the measurement are specified in SEMATECH [17], and the measurement was performed using the equipment that complies with them. QMS is a mass spectrometer that uses an ion separation quadrupole and mainly used for gas analysis. In this measurement, it is used for the purpose of calculation the flow-rate of process exhaust gas by measuring the concentration of the injected tracer gas and calculating the dilution ratio and mass analyzer with a minimum specification of 0 to 100 amu should be used to include both the composition of the exhaust gas and the mass range of the tracer gas. Additionally, to maintain the vacuum pressure, vacuum pump and pressure gauge were installed and used together. FT-IR is an analysis device that enables qualitative and quantitative analysis by measuring the number of infrared rays absorbed by sample, and it is used for the purpose of measuring the concentration of the target substance contained in the exhaust gas from the process. The gas cell installed in the FT-IR used standardized cells and vacuum-only fittings and it is accurately placed on the holder by attaching the window made of KBr, ZnSe, or Ge/ZnSe material through which infrared light passes. The gas cell suitable for the type and concentration of the target gas should be applied, and length for each type of scrubber as mentioned Table S3.

Figure 1. Diagram of measuring device installation for measuring DRE.

This measurement guideline for estimating DRE, Use rate of gas (Ui) and By-product emission factor (Bby-product, i) applied nitrous oxide (N2O), hydrofluorocarbon (HFCs), perfluorocarbons (PFCs), sulfur hexafluoride (SF6), and nitrogen trifluoride (NF3) among semiconductor and display manufacturing process emissions as target substance in order to apply the GHG emissions calculation presented in the guidelines for reporting and certification of GHG emissions trading. Hydrofluorocarbons included HFC-23 (CHF3), HFC-32 (CH2F2), and perfluorocarbons included PFC-14 (CF4), PFC-116 (C2F6), PFC-218 (C3F8) and PFC-c318 (c-C4F8) and in this measurement method, the detection limit (DL) for each gas cell of FT-IR is in accordance with Table S2.

Interfering substances that affect the measurement’s results in the process included moisture are spectrum interference. In order to eliminate this interference, the gas sample must be heated to 100℃ or more before the gas sample flows into the analysis device to minimize the inflow of moisture and substances causing interference due to overlapping with the reference spectrum region of the target gas to be measure should be reset to a spectrum that does not overlap to minimize interference.

The tracer gas injected into the pipeline to calculate the flow-rate of process exhaust gas is chemically stable and well mixed and diffused with the atmosphere, and tracer gas applied and used one of helium (He), neon (Ne), argon (Ar), krypton (Kr), and xenon (Xe) gases which are inert gases used for measuring the volume flow-rate of GHG emissions from the manufacturing process. The injecting position of the tracer gas is at the rear end of the pump in manufacturing process and the distance between the inlet of the tracer gas and the inlet of the scrubber should be installed at least 1 m (POU scrubber) or 5 m (house scrubber) so that the tracer gas can be sufficiently mixed. Before the measurement, the whole part of the of the measuring instrument should be inspected. In particular, it is necessary to check whether gas is leaking, turn on the power according to the order, and adjust the sample’s standby flow rate and other conditions according to the manual. And then when steady-state is reached, QMS and FT-IR instruments should calibrate for accurate measurement. Finally, on-site at the manufacturing facilities during normal operation, QMS and FT-IR are operated to estimate volume-flow rate and concentration of the target gas continuously for 1 hour to calculate DRE, Use rate of gas (Ui) and By-product emission factors (Bby-product, i). Measurements are calculated at each scrubber in semiconductor & display facilities.

Point 3: How about validation for improvement for section 2.2 and 2.3?

Response 3: Thank you for your thoughtful comment. We have now added the session 2.3.3 and 2.4.3 about validation of improved measurement (Use rate of gas and By-product emission factor).

(Page 12 – Materials and Methods)

2.3.3. Validation of improved measurement

The validation the use of rate (Ui) was confirmed simultaneously through the validity evaluation of the previously mentioned DRE measurement. it was applied equally to Use rate of gas (Ui) and By-product Emission Factor (Bby-product, i), and the results mentioned in 2.2.4 were verified.

2.4.3. Validation of improved measurement

The validation of By-product Emission Factor (Bby-product, i) was also conducted simultaneously through the validity evaluation of the DRE measurement mentioned above. it was applied equally to Use rate of gas (Ui) and By-product Emission Factor (Bby-product, i), and the results mentioned in 2.2.4 were verified.

Point 4: Figure 6 is quite blur.

Response 4: Thanks for catching our mistake. We have now revised Figure 6 to be clearer.

Figure 6. Diagram for measuring Destruction Removal Efficiency (DRE)

Point 5: Table 0 cannot be found.

Response 5: We are sorry for the omission. We have now revised from Table 0 to Table 7 and Table 13.

(Page 13 – 3.1.2. Results of measurement)

In order to confirm the optimal conditions for each gas of the POU scrubber, DRE for each volume-flow rate and consumption of plasma energy was measured. When the treatment volume-flow rate of the POU scrubber was increased to 300 SLM, it was confirmed that the DRE was decreased to some extent. In particular, in case of NF3, it is necessary to increase the energy consumption for optimal condition of DRE and the reason for setting 300 SLM is the applicable optimal processing capacity in the display sectors. The results of measurement were shown in Table 7.

(Page 16– 3.4. Results of DRE, Use rate of gas (Ui) and By-product Emission Factors (Bby-product, i) measurement)

According to the results of By-product Emission Factors (Bby-product, i) measurement, comparison was performed with 06’ IPCC G/L and 19’ IPCC Refinement to the ’06 IPCC G/L for National Greenhouse Gas Inventories as shown in Table 13.

Point 6: Results for section 3.2 and 3.3 need more explanation.

Point 7: Suggest to have one section specially for comparison (instead of putting all in the conclusion).

Response 6, 7: Taking your advice, we have now moved the tables from conclusions to results and then additional session 3.4 were created to organize the contents for comparison of session 3.1, 3.2, 3.3.

(Page 15, 16 – Results)

3.4. Results of DRE, Use rate of gas (Ui) and By-product Emission factors (Bby-product, i) measurement

According to the results of DRE measurement, comparison was performed with the 06’ IPCC G/L and 19’ IPCC Refinement to the ’06 IPCC G/L for National Greenhouse Gas Inventories as following Table 11. It was confirmed that DRE showed a large difference by volume-flow rate, gas type, and energy consumption in manufacturing process. As a result, since DRE coefficients for each gas presented in the IPCC G/L considered and presented only the type of gas, and it was necessary to develop coefficients considering various facility conditions in order to apply DRE suitable for the site.

Point 8: For conclusion, suggest to provide more detailed recommendation based on the limitation of current study.

Response 8: Thank you for your comment. We have now added phrases.

(Page 16, 17 - Conclusion)

In summary, we presented accurate measurement methods of DRE, Use rate of gas (Ui) and By-product emission factors (Bby-product, i) which are characteristic parameters for estimating GHGs emission in the semiconductor and display industry and fluorine-based GHGs used in the electronics manufacturing processes were measured at Korea’s various facilities. As a result, it was confirmed GHG emissions and national GHG emissions are overestimated compared to the reduction and emission factor of ’06 IPCC G/L and ’19 IPCC G/L refinement currently applied to calculate GHG emissions in Republic of Korea. Through this study, based on the established GHGs emission calculation characteristic parameters in the semiconductor & display industry, the need for re-establishment considering the facility’s applicability of the various coefficients presented in EPA 40 U.S. Code of Federal Regulations (CFR) part 98; Subpart I, ’06 IPCC G/L and ’19 IPCC refinement to the ’06 IPCC G/L for National Greenhouse Gas Inventories was confirmed, and through the measurement of Destruction Removal Efficiency (DRE), Use rate of gas (Ui) and By-product Emission Factors (Bby-product, i) for GHGs that used in the manufacturing process in semiconductor & display industry but not presented in IPCC G/L, the need for additional study was reminded to prevent overestimation of GHG emissions in the Republic of Korea’s semiconductor & display industry. Furthermore, we believe that reduction and emission factors unique to the Republic of Korea, which are more accurately established through additional measurement of GHGs used in semiconductor and display manufacturing processes in Korea and research on characteristic parameters, can significantly contribute to the reduction of national GHG emissions and it can be provided as a good reference for countries that need to develop country-specific reduction & emission factors.

Reviewer 3 Report

This paper aims to propose new parameters for GHG emission estimation for the Korean semi-conductor and display industry, as an alternative to the average parameters used internationally.

In its current form, it is very difficult to see the scientific contribution of the manuscript.

The introduction section is very direct and very specific. It is difficult to understand the context. I would recommend you to elaborate more on previous research to contextualize the study.

There is practically no literature review at present, and no discussion on the results in relation to extant literature. In the cases where references are actually used, they are often presented in anonymous lumps.

The conditions for the experiments are not clear enough. It seems like the results are based on measurements on a single site? Is that enough to ensure that the parameters can be used across the whole industry?

The overall structure of the manuscript is quite confusing. There are descriptions of methodology in the results section and vice versa. The conclusions section consists mainly of comparisons of results from the study to standard values for the industry. While such a comparison could be relevant for the study, it would rather belong in the results section.

How much better (more representative) are the proposed factors for the intended application than the standard values? To what extent is the method possible to generalize?

Should be improved and checked by a native English speaker.

Author Response

Response to Reviewer 3 Comments

Reviewer #3

Reviewer’s Comment: This paper aims to propose new parameters for GHG emission estimation for the Korean semi-conductor and display industry, as an alternative to the average parameters used internationally. In its current form, it is very difficult to see the scientific contribution of the manuscript. The introduction section is very direct and very specific. It is difficult to understand the context. I would recommend you to elaborate more on previous research to contextualize the study. There is practically no literature review at present, and no discussion on the results in relation to extant literature. In the cases where references are actually used, they are often presented in anonymous lumps. The conditions for the experiments are not clear enough. It seems like the results are based on measurements on a single site? Is that enough to ensure that the parameters can be used across the whole industry? The overall structure of the manuscript is quite confusing. There are descriptions of methodology in the results section and vice versa. The conclusions section consists mainly of comparisons of results from the study to standard values for the industry. While such a comparison could be relevant for the study, it would rather belong in the results section. How much better (more representative) are the proposed factors for the intended application than the standard values? To what extent is the method possible to generalize?

Response : We appreciate your comments.

Specific Comments

Point 1: The introduction section is very direct and very specific. It is difficult to understand the context. I would recommend you to elaborate more on previous research to contextualize the study.

Point 2: There is practically no literature review at present, and no discussion on the results in relation to extant literature. In the cases where references are actually used, they are often presented in anonymous lumps.

Response 1,2: Thank you for your comment. We have now revised introduction and added references.

(Page 1,2 – introduction)

The semiconductor and display industries in Republic of Korea are continuously increasing greenhouse gas (GHG) emissions due to the rapid growth in production volume [1-7]. 60-70 percent of GHG emissions from the semiconductor & display industries are indirect emissions released through electricity consumption but it is still insufficient to contribute to GHG reduction for carbon neutrality by improving process efficiency [7, 8].

There is a lack of related studies for GHG assessment in Korea. With these issues, various industries used the default emission factor presented in the Intergovernmental Panel on Climate Change (IPCC) guidelines for national GHG inventories to date. The IPCC currently recommended applying country-specific reduction and emission factors rather than default values [9]. When the country-specific emission factors for each industry that have been studied recently are developed and compared with the previously applied IPCC default values, the results showing significantly difference were confirmed. Comparing these results, it reflected that the domestic semiconductor and display industries are overestimating GHG emissions by using IPCC default factors. Therefore, recognizing the fact calculation of GHG emissions is being overestimated, various studies are being conducted to develop the country-specific emission factors [9-13]. Through these efforts, the determination of appropriate GHG emission factors suitable for nation’s conditions will be essential data for domestic GHG forecasting and reduction strategy establishment. Currently, process emissions from fluorinated gases in the domestic semiconductor and display industries are calculated based on the ’06 IPCC guidelines, and emission factors applied have also been based on the ’06 IPCC default values [14, 15]. However, reduction and emission factors presented in the IPCC guidelines have quite conservative and non-reflected the latest advanced abatement technologies in the domestic semiconductor & display industries. As a result, GHG emissions from domestic semiconductor & display facilities and national GHG emissions are being overestimated. Accordingly, developing site-specific emission factors at the workplace level through direct measurement of GHG emission calculation characteristic parameters is necessary, reduction of GHG emissions and carbon neutrality could be realized. Nevertheless, there is currently insufficient research on how to directly estimate the characteristic parameters of GHG emissions calculation, but through measurement research on the characteristic parameters of GHG emission calculation, we proposed clear measurement methods for securing the reliability of GHG emissions over facilities data in the semiconductor & display sectors and Destruction Removal Efficiency (DRE), Use rate of gas (Ui) and By-product Emission Factor (Bbyproduct, i).

  1. Eui-Chan Jeon, Soojeong Myeong, Jae-Whan Sa, Jinsu Kim, Jae-Huk Jeong, Greenhouse gas emission factor development for coal-fired power plants in Korea, Applied Energy, Vol 87, 1, 205-210, 2010
  2. Lee, J. Kim, J. Lee & E.-C. Jeon, Development of CO2 emission factors from a large circulating fluidized bed boiler, Energy Sources, Part A: Recovery, Utilization and Environmental Effects, Vol.38, 9, 1262-1268, 2016
  1. Changsang Cho, Seongmin Kang, Minwook Kim, Yoonjung Hong, and Eui-chan Jeon, Uncertainty Analysis for the CH4 Emission Factor of Thermal Power Plant by Monte Carlo Simulation, Sustainability (Switzerland), Vol. 10, 10, 2018
  2. Seongmin Kang, Seonghum Cho, Joonyoung Roh, and Eui-chan Jeon, Analysis of Main Factors for CH4 Emission Factor Development in Manufacturing Industries and Construction Sector, Energies, Vol. 13, 5, 2020
  3. Seongmin Kang, Seong-Dong Kim, and Eui-chan Jeon, Emission Characteristics of Ammonia at Bituminous Coal Power Plant, Energies, Vol. 13, 7, 2020

Point 3: The conditions for the experiments are not clear enough. It seems like the results are based on measurements on a single site? Is that enough to ensure that the parameters can be used across the whole industry?

Response 3: Thank you for your suggestion. We have now added session in Materials and Methods. In session 2.1, the measurement methods and conditions were described in detail.

(Page 3,4 – Materials and Methods)

2.1. Measurement methods anc conditions of DRE, Use rate of gas (Ui), By-product emission factor (Bby-product, i)

Calculation of DRE, Use rate of gas (Ui), By-product emission factor (Bby-product, i) is measured on-site at the semiconductor & display manufacturing facilities during normal operation. Considering the on-site circumstances and process gas being used, the volume-flowrate of the process gas entering and emitting from the abatement equipment is measured using quadrupole mass spectrometer (QMS), and the concentration of the targeted FCs or N2O, is measured using fourier transform infra-red spectrometer (FT-IR) (Figure 1).

The QMS and FT-IR equipment regulations used for the measurement are specified in SEMATECH [17], and the measurement was performed using the equipment that complies with them. QMS is a mass spectrometer that uses an ion separation quadrupole and mainly used for gas analysis. In this measurement, it is used for the purpose of calculation the flow-rate of process exhaust gas by measuring the concentration of the injected tracer gas and calculating the dilution ratio and mass analyzer with a minimum specification of 0 to 100 amu should be used to include both the composition of the exhaust gas and the mass range of the tracer gas. Additionally, to maintain the vacuum pressure, vacuum pump and pressure gauge were installed and used together. FT-IR is an analysis device that enables qualitative and quantitative analysis by measuring the number of infrared rays absorbed by sample, and it is used for the purpose of measuring the concentration of the target substance contained in the exhaust gas from the process. The gas cell installed in the FT-IR used standardized cells and vacuum-only fittings and it is accurately placed on the holder by attaching the window made of KBr, ZnSe, or Ge/ZnSe material through which infrared light passes. The gas cell suitable for the type and concentration of the target gas should be applied, and length for each type of scrubber as mentioned Table S3.

Figure 1. Diagram of measuring device installation for measuring DRE.

This measurement guideline for estimating DRE, Use rate of gas (Ui) and By-product emission factor (Bby-product, i) applied nitrous oxide (N2O), hydrofluorocarbon (HFCs), perfluorocarbons (PFCs), sulfur hexafluoride (SF6), and nitrogen trifluoride (NF3) among semiconductor and display manufacturing process emissions as target substance in order to apply the GHG emissions calculation presented in the guidelines for reporting and certification of GHG emissions trading. Hydrofluorocarbons included HFC-23 (CHF3), HFC-32 (CH2F2), and perfluorocarbons included PFC-14 (CF4), PFC-116 (C2F6), PFC-218 (C3F8) and PFC-c318 (c-C4F8) and in this measurement method, the detection limit (DL) for each gas cell of FT-IR is in accordance with Table S2.

Interfering substances that affect the measurement’s results in the process included moisture are spectrum interference. In order to eliminate this interference, the gas sample must be heated to 100℃ or more before the gas sample flows into the analysis device to minimize the inflow of moisture and substances causing interference due to overlapping with the reference spectrum region of the target gas to be measure should be reset to a spectrum that does not overlap to minimize interference.

The tracer gas injected into the pipeline to calculate the flow-rate of process exhaust gas is chemically stable and well mixed and diffused with the atmosphere, and tracer gas applied and used one of helium (He), neon (Ne), argon (Ar), krypton (Kr), and xenon (Xe) gases which are inert gases used for measuring the volume flow-rate of GHG emissions from the manufacturing process. The injecting position of the tracer gas is at the rear end of the pump in manufacturing process and the distance between the inlet of the tracer gas and the inlet of the scrubber should be installed at least 1 m (POU scrubber) or 5 m (house scrubber) so that the tracer gas can be sufficiently mixed. Before the measurement, the whole part of the of the measuring instrument should be inspected. In particular, it is necessary to check whether gas is leaking, turn on the power according to the order, and adjust the sample’s standby flow rate and other conditions according to the manual. And then when steady-state is reached, QMS and FT-IR instruments should calibrate for accurate measurement. Finally, on-site at the manufacturing facilities during normal operation, QMS and FT-IR are operated to estimate volume-flow rate and concentration of the target gas continuously for 1 hour to calculate DRE, Use rate of gas (Ui) and By-product emission factors (Bby-product, i). Measurements are calculated at each scrubber in semiconductor & display facilities.

Point 4: The overall structure of the manuscript is quite confusing. There are descriptions of methodology in the results section and vice versa. The conclusions section consists mainly of comparisons of results from the study to standard values for the industry.

Point 5: While such a comparison could be relevant for the study, it would rather belong in the results section. How much better (more representative) are the proposed factors for the intended application than the standard values? To what extent is the method possible to generalize?

Response 4, 5: Thank you for your thoughtful comment. We have now moved the tables from conclusions to results and then additional session 3.4 were created to organize the contents for comparison of session 3.1, 3.2, 3.3 (Compared with default factors presented ’06 and ‘19 IPCC G/L).

(Page 15~16 – Results)

3.4. Results of DRE, Use rate of gas (Ui) and By-product Emission factors (Bby-product, i) measurement

According to the results of DRE measurement, comparison was performed with the 06’ IPCC G/L and 19’ IPCC Refinement to the ’06 IPCC G/L for National Greenhouse Gas Inventories as following Table 11. It was confirmed that DRE showed a large difference by volume-flow rate, gas type, and energy consumption in manufacturing process. As a result, since DRE coefficients for each gas presented in the IPCC G/L considered and presented only the type of gas, and it was necessary to develop coefficients considering various facility conditions in order to apply DRE suitable for the site.

Round 2

Reviewer 2 Report

Major revision has been made by the authors. It is suggested to mention the validation of the results in section 3 as well. So far, it is only mentioned in section 2 as the methodology.

Please proofread the manuscript. Check the word formatting also (line 418, capitalized first word).